# Adaptive Regularization for Class-Incremental Learning

**Elif Ceren Gok Yildirim**[1]  **Murat Onur Yildirim**[1]  **Mert Kilickaya**[1]  **Joaquin Vanschoren**[1]

[1]Automated Machine Learning Group, Eindhoven University of Technology

**Abstract**  Class-Incremental Learning updates a deep classifier with new categories while maintaining the previously observed class accuracy. Regularizing the neural network weights is a common method to prevent forgetting previously learned classes while learning novel ones. However, existing regularizers use a constant magnitude throughout the learning sessions, which may not reflect the varying levels of difficulty of the tasks encountered during incremental learning. This study investigates the necessity of adaptive regularization in Class-Incremental Learning, which dynamically adjusts the regularization strength according to the complexity of the task at hand. We propose a Bayesian Optimization-based approach to automatically determine the optimal regularization magnitude for each learning task. Our experiments on two datasets via two regularizers demonstrate the importance of adaptive regularization for achieving accurate and less forgetful visual incremental learning.

## 1 Motivation

This paper focuses on Class-Incremental Learning of deep neural network representations (Masana et al., 2020; De Lange et al., 2021). Unlike standard batch learning, which requires access to data from all categories simultaneously, Class-Incremental Learning can update a pre-trained deep classifier with new categories without revisiting old data. This allows for more efficient learning and avoids the need to store large amounts of old data.

While Class-Incremental Learning enables expanding a classifier without retraining, it often results in a significant cost known as *catastrophic forgetting*. This occurs when the deep learner sacrifices accuracy on previously seen classes to learn new ones. Two major approaches have been explored to address this issue: regularization (Kirkpatrick et al., 2017; Li and Hoiem, 2017) and replay (Lopez-Paz and Ranzato, 2017). Regularization prevents abrupt shifts in the neural network weights, while replay stores a few exemplars per-class in memory and replays them during new learning increments. This paper focuses on regularization, as it is simple yet effective.

Regularization-based approaches to address catastrophic forgetting can be categorized into two main types: prior-based and distillation-based. Prior-based methods predict the importance of each weight and ensure their stability across learning sessions using techniques such as the Fisher matrix (Le Cam, 2012). Examples of prior-based methods include EwC (Kirkpatrick et al., 2017), SI (Zenke et al., 2017), RWalk (Chaudhry et al., 2018a), and IMM (Lee et al., 2017). In contrast, distillation-based approaches work in a teacher-student setting and align the output of the previous teacher model with the current student model. Examples of distillation-based methods include LwF (Li and Hoiem, 2017), iCaRL (Rebuffi et al., 2017), BiC (Wu et al., 2019), and COIL (Zhou et al., 2021b). While these methods improve performance, they use a fixed regularization magnitude throughout learning and focus solely on *how* to regularize.

This paper addresses the issue of *how much* to regularize in Class-Incremental Learning for the first time. We explore whether adaptation is necessary for optimal performance, treating the regularization magnitude as a latent variable that should be adjusted based on the current state of the learner and the complexity of the task (see Figure 1). We use Bayesian Optimization to predict the best regularization strength per task. Our experiments on CIFAR-100 and MiniImageNet demonstrate that adapting to the current task results in significant improvement, even in these simple scenarios.

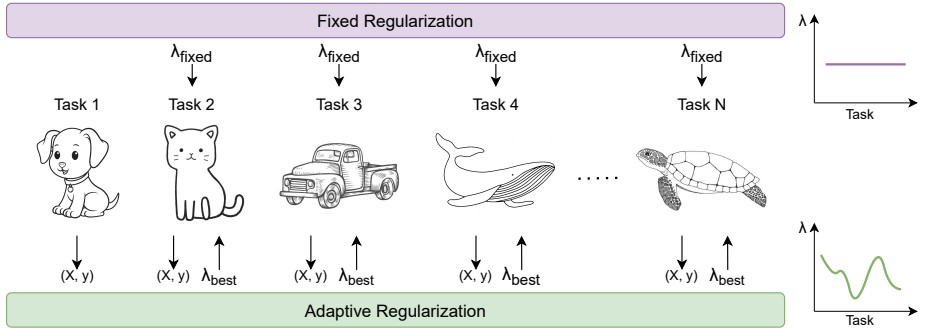

Figure 1: A comparison of fixed *vs.* adaptive regularization (ours). In this work, we explore the potential of tuning regularization per-learning task, allowing to learn adaptively.

## 2 Related Work

**Class-Incremental Learning.** Class-Incremental Learning updates a deep classifier with sequentially arriving data, usually with mutually exclusive categories (Masana et al., 2020; De Lange et al., 2021; Wang et al., 2023; Zhou et al., 2023; Kilickaya et al., 2023). However, when novel data arrives, previous training data becomes unavailable, leading to catastrophic forgetting. To mitigate this, researchers have developed two main approaches: replay methods, which store a subset of training data to rehearse during learning (Lopez-Paz and Ranzato, 2017; Chaudhry et al., 2018b; Aljundi et al., 2019; Ostapenko et al., 2019; Xiang et al., 2019), and regularization-based approaches, which stabilize important parameters or distill previous knowledge into the model (Kirkpatrick et al., 2017; Chaudhry et al., 2018a; Zenke et al., 2017; Lee et al., 2017; Li and Hoiem, 2017; Rebuffi et al., 2017; Wu et al., 2019; Zhou et al., 2021b).

However, assuming a constant amount of regularization throughout learning sessions is unnatural, since learning unfamiliar objects requires more plasticity than learning familiar ones. To address this issue, we propose a regularization method in which the regularization magnitude is a function of time and is automatically tuned using Bayesian Optimization (Turner et al., 2021). We evaluate the generality of our approach by choosing EWC (Kirkpatrick et al., 2017) as the prior-based regularizer and LwF (Li and Hoiem, 2017) as the distillation-based regularizer.

**Hyper-Parameter Optimization.** Hyper-Parameter Optimization (HPO) aims to optimize the hyper-parameters of a given deep learning model, including the learning rate, layer size, or balance of different loss functions. In this paper, our focus is on balancing the contribution of standard classification and the regularization loss. To tackle the HPO problem, complex techniques such as bi-level optimization (Franceschi et al., 2018) or gradient-based optimization (Baydin et al., 2018) have been proposed. Bi-level optimizers alternate between optimizing neural network weights and tuning the hyper-parameters, while gradient-based methods treat the entire network weights as a hyper-parameter to be updated.

However, in this work, we propose to use Bayesian Optimization (Snoek et al., 2012) due to its simplicity and effectiveness. In summary, this paper makes the following contributions:

I. In this paper, for the first time, we raise the important issue of adaptive regularization in class-incremental learning.

II. We propose to predict the regularization magnitude conditioned on the state of the deep learner and the current learning task via Bayesian Optimization.

III. Through large-scale experiments on well-established benchmarks, we show that learning adaptively yields significant performance improvements, in terms of increasing accuracy while reducing forgetting.

## 3 Empirical Method

**Overview.** Class-incremental learning involves updating a neural network with new data as it comes in. Specifically, the learner receives a sequence of learning tasks $\mathcal{T}_{1:t} = (\mathcal{T}_1, \mathcal{T}_2, ..., \mathcal{T}_t)$, each with a corresponding dataset $\mathcal{D}_\mathcal{T} = (x_{i,t}, y_{i,t})^{n_t}$ consisting of $n_t$ instances per task. Each input pair $x_{i,t}, y_{i,t} \in \mathcal{X}t \times \mathcal{Y}t$ is sampled from an unknown distribution. It's important to note that the learning tasks are mutually exclusive, i.e., $\mathcal{Y}_{t-1} \cap \mathcal{Y}_t = \emptyset$.

When a new learning task arrives, a deep convolutional network is optimized to embed the input instance into the classifier space $f_\Theta : \mathcal{X}_t \rightarrow \mathcal{Y}_t$, where $\Theta$ represents the parameters of the learner. To improve clarity, we will drop the subscript moving forward.

The incremental learner has two goals: to effectively learn the current task (*plasticity*) while retaining performance on all previous tasks (*stability*). This is accomplished by optimizing the following function:

$$\mathcal{L} = CE(f(x_{i,t}), y_{i,t}) + \lambda \cdot Reg(\Theta) \tag{1}$$

Here, $CE(\cdot)$ represents the standard Cross-Entropy used in classification, and $Reg(\cdot)$ is a regularization term that penalizes abrupt changes in the neural network weights (Li and Hoiem, 2017; Kirkpatrick et al., 2017).

**Regularization Constancy Assumption.** The scalar parameter $\lambda$ balances the contribution of the classification and regularization loss functions. A large value of $\lambda$ ensures minimal weight updates, which can sacrifice learning on the current task. Conversely, a small $\lambda$ yields good performance on the current task but may sacrifice performance on previous tasks, exacerbating catastrophic forgetting. As a common practice, $\lambda$ is set to a fixed scalar value throughout all incremental learning sessions, such that $\lambda_{t-1} = \lambda_t$ for all $t \in \mathcal{T}_{1:t}$.

We hypothesize that the assumption of *regularization constancy* is unrealistic for building accurate lifelong learning machines. Our reasoning is twofold:

- **Low Plasticity and High Stability:** The incremental learner may encounter a novel object that is highly familiar with the previously learned tasks. For example, it may encounter the category *dog* after observing many other animal categories, such as *cat, cow, bird*. In this case, the learner does not need to be too plastic, as it can quickly transfer knowledge from the previous tasks where it is similar to the human learning process and refered as *low road transfer* concept (Perkins et al., 1992). Hence, no drastic updates to the learned filters are necessary.

- **High Plasticity and Low Stability:** The incremental learner may encounter a novel object that is highly unfamiliar with the previous tasks. For example, it may encounter the category *car* after observing many other animal categories, such as *cat, cow, bird*. In this case, the learner requires high plasticity to learn about the novel object with never-before-seen parts, such as wheels (Perkins et al., 1992).

**Adaptive Regularization.** This paper argues that the assumption of regularization constancy is unrealistic and proposes an alternative approach: adaptive regularization. In this approach, the regularization magnitude is a function that consists set of incremental tasks, conditioned on the current learning task and all previous tasks. Formally, we define $\lambda(t) = \lambda_1, \lambda_2, \ldots, \lambda_{t-1}, \lambda_t$, where $\lambda_t$ is predicted by minimizing the following optimization problem:

$$\lambda^* = \arg\min_\lambda \mathcal{L}(\Theta; \lambda, V_t) = \arg\min_\lambda \sum_{i=1}^{|V_t|} \left[ CE(f(x_{i,t}; \Theta), y_{i,t}) + \lambda \cdot Reg(\Theta) \right] \; where \; \lambda \geq 0 \tag{2}$$

Here, $V_t$ is the subset of the validation set for the current task and previous tasks, and $\mathcal{L}(\Theta; \lambda, V_t)$ is the loss function with the regularization coefficient $\lambda$ determined by solving the optimization problem. This approach allows us to automatically adjust the regularization strength according to the specific learning task based on the given loss function which let the model find the degree of difficulty itself, avoiding the unrealistic assumption of a fixed regularization strength throughout the learning process.

In what follows, we describe the regularization as well as the optimization objectives adopted in our work.

### 3.1 Regularization

To regularize the weights of the backbone, we experimented with two popular, well-established techniques: LwF (Li and Hoiem, 2017) and EWC (Kirkpatrick et al., 2017).

**LwF.** Learning-without-Forgetting (Li and Hoiem, 2017) is a knowledge-distillation approach where the teacher branch is the model from the previous task, and the student branch is the current model. The aim is to match the activations of the teacher and student branches, either at the feature or logit layer. We found that logit-based distillation yielded better performance. Formally, LwF minimizes the following objective:

$$Reg(\Theta) = KL(f(x_{i,t}), f'(x_{i,t})) \tag{3}$$

where $f'$ is the model from the previous learning step, and $KL(p_1, p_2)$ is the KL-divergence between two probability distributions $p_1$ and $p_2$.

**EWC.** Elastic Weight Consolidation (Kirkpatrick et al., 2017) can be viewed as a weighted regularization approach. The authors argue that not all weights contribute equally to learning a new task. Thus, they estimate the importance of each weight in minimizing the classification loss for the current task: $Reg(\Theta) = ||\mathcal{F}(\Theta - \Theta')||$, where $\Theta'$ is the model weights from the previous learning step, $\mathcal{F}$ is the Fisher matrix of the same size as the weight matrices $\Theta$, re-weighting the contributions of each weight to stabilize the important neurons per task.

### 3.2 Bayesian Optimization via Parzen Estimator

We optimize the equation 2 using multivariate tree-structured parzen estimator (TPE) (Bergstra et al., 2011), thanks to its high-performance and accuracy. TPE builds a conditional probability tree that maps hyperparameters to their respective model performances. Then it can be used to guide a search algorithm to find the optimal set of hyperparameters for the given model. In this study, Bayesian Optimization is utilized as a search algorithm where it searches within the provided range for regularization strength and selects the best value. Specifically, we build upon the implementation from the RayTune library (Liaw et al., 2018).

The estimator is trained on the validation set accumulated across incremental learning sessions. We store a small subset of 20% of the validation data from each incremental learning step in the memory. The search space for $\lambda$ is set to $[1, 10^5]$ for EWC and we uniformly sampled 20 configurations. Similarly, the search space for $\lambda$ is set to $[1, 50]$ for LwF and uniformly sampled 15 configurations. These ranges were determined based on the sensitivity analysis which is experimented on CIFAR 10 and given on the Appendix in Figure 4 and Figure 5. As for the baselines, we used one naive and one strong approaches: For the naive one, we kept $\lambda$ fixed and set it to 1 which assumes the current task and previous tasks should have equal learning importance; for the strong baseline, we adopt cosine annealing where $\lambda$ starts from the maximum value of defined range and get decreased at each incremental step.

**Implementation Details.** We implement all the methods in PyTorch (Paszke et al., 2019). We use ResNet-32 as the backbone (He et al., 2016). We rely on the PyCIL library (Zhou et al., 2021a) for the regularization objectives EWC (Kirkpatrick et al., 2017) and LwF (Li and Hoiem, 2017), and use all the hyperparameters as is without further tuning. We set number of epochs to 100 for each configuration but used the Asynchronous Successive Halving (Li et al., 2018) scheduler for a more efficient search. We set an initial learning rate of 0.1 which is decayed by factor 0.1 at epoch 60 for the first task and 70 for the rest of the tasks. We use SGD optimizer with momentum parameter set to 0.9 and weight decay set to $5e^4$ for the first task and $2e^4$ for the rest of the tasks. The batch size is set to 128. We run experiments on three different seeds $(2, 1993, 2022)$ and report their average.

## 4 Experimental Setup

**Datasets.** In this paper, we experiment with **Split-CIFAR100** (Krizhevsky et al., 2009) and **Split-MiniImageNet** (Vinyals et al., 2016). Each dataset exhibits objects from 100 different categories, such as bird, snake and spider. We train all the models with 10 tasks, with 10 classes within each learning task on both CIFAR100 and MiniImageNet. Both datasets have 5000 training, and 1000 testing color images per learning task, each with $32 \times 32$ and $64 \times 64$ resolution for CIFAR100 and MiniImageNet respectively.

**Metrics.** We resort to the standard metrics for evaluation, accuracy (ACC) which measures the final accuracy averaged over all tasks, and backward transfer (BWT) which measures the average accuracy change of each task after learning new tasks. Formally for accuracy:

$$ACC = \frac{1}{T} \sum_{i=1}^{T} A_{T,i},$$
(4)

and for backward transferability:

$$BWT = \frac{1}{T-1} \sum_{i=1}^{T-1} (A_{T,i} - A_{i,i})$$
(5)

where $A_{T,i}$ represents the testing accuracy of task $T$ after learning task $i$. In both cases, higher values indicate better performance.

## 5 Analysis

In this section, we address the following three research questions:

- **RQ1**: Does adaptive regularization improve backward transfer?

- **RQ2**: To what extent does adaptive regularization improve incremental learning accuracy per-task?

- **RQ3:** How does the predicted $\lambda$ values differ across different incremental learning steps?

### 5.1 RQ1: Does Adaptive Regularization Improve Backward Transfer?

First, we investigate the backward transfer performance of fixed *vs.* cosine annealing *vs.* adaptive regularization. The results are presented in Table 1.

| CIFAR-100 | | | | |
|---|---|---|---|---|
| | EWC | | LwF | |
| Method | ACC | BWT | ACC | BWT |
| Fixed | $9.11 \pm 0.08$ | $-86.16 \pm 1.52$ | $18.91 \pm 0.13$ | $-56.58 \pm 10.14$ |
| Cosine | $19.94 \pm 1.47$ | $-6.16 \pm 1.07$ | $15.48 \pm 1.28$ | $-32.03 \pm 8.12$ |
| Adaptive (ours) | $23.32 \pm 0.87$ | $-8.03 \pm 4.76$ | $27.29 \pm 1.64$ | $-25.24 \pm 3.63$ |
| $\Delta_{fixed}$ | 14.21 | 78.13 | 8.38 | 31.34 |
| $\Delta_{cosine}$ | 3.38 | $-1.87$ | 11.81 | 6.79 |
| **MiniImageNet** | | | | |
| | EWC | | LwF | |
| Method | ACC | BWT | ACC | BWT |
| Fixed | $8.84 \pm 0.22$ | $-84.87 \pm 1.25$ | $15.14 \pm 0.48$ | $-64.20 \pm 8.11$ |
| Cosine | $17.50 \pm 1.49$ | $-8.21 \pm 1.69$ | $21.67 \pm 0.75$ | $-42.26 \pm 6.93$ |
| Adaptive (ours) | $21.01 \pm 0.78$ | $-6.94 \pm 0.79$ | $28.23 \pm 1.72$ | $-27.09 \pm 4.79$ |
| $\Delta_{fixed}$ | 12.17 | 77.93 | 13.09 | 37.11 |
| $\Delta_{cosine}$ | 3.51 | 1.27 | 6.56 | 15.17 |

Table 1: Final task accuracy and backward transfer scores of adaptive *vs.* cosine annealing *vs.* fixed regularization on CIFAR100 and MiniImageNet. Regularizing adaptively outperforms fixed regularization by a large margin, in both ACC and BWT.

As can be seen from Table 1, adaptive regularization outperforms on both CIFAR100 and MiniImageNet. In CIFAR100 experiments, adaptivity has a more significant improvement on the accuracy of EWC. Interestingly, a strong baseline, namely cosine annealing, exhibits inferior incremental accuracy compared to a simple, fixed baseline; nevertheless, it still enhances the backward transfer of the model. In MiniImageNet experiments, on the other hand, LwF improved slightly better than EWC in terms of accuracy score.

We also observed that BWT performance significantly improved on EWC compared to LwF with both cosine annealing and adaptive approach which signals EWC is more prone to forgetting, thus requiring better adaptation to the current learning task.

### 5.2 RQ2: To What Extent Adaptive Regularization Improve Incremental Learning Per-Task?

In our second analysis, we compare the fixed, cosine and adaptive versions of EWC and LwF. Results are presented in Figure 2.

As can be seen, regardless of the method, adaptivity leads to a significant increase in top-1 accuracy, sometimes leading up to 20% for EWC and 12% for LwF on CIFAR100, while up to 14% for EWC and 15% for LwF on MiniImageNet compared to fixed baseline. Furthermore, the adaptive approach even improves over a strong baseline by reaching up to 3% for EWC and 12% for LwF on CIFAR100, while up to 4% for EWC and 7% for LwF on MiniImageNet. This confirms our hypothesis that regularization-based class-incremental learners can benefit a lot from learning to adapt to the current task. Taken together, the results demonstrate the superior performance of the proposed adaptive strategy over both baseline methods.

### 5.3 RQ3: How Does the Predicted $\lambda$ Values Differ Across Incremental Learning Steps?

In our last analysis, we present per-incremental learning task $\lambda$ values in Figure 3.

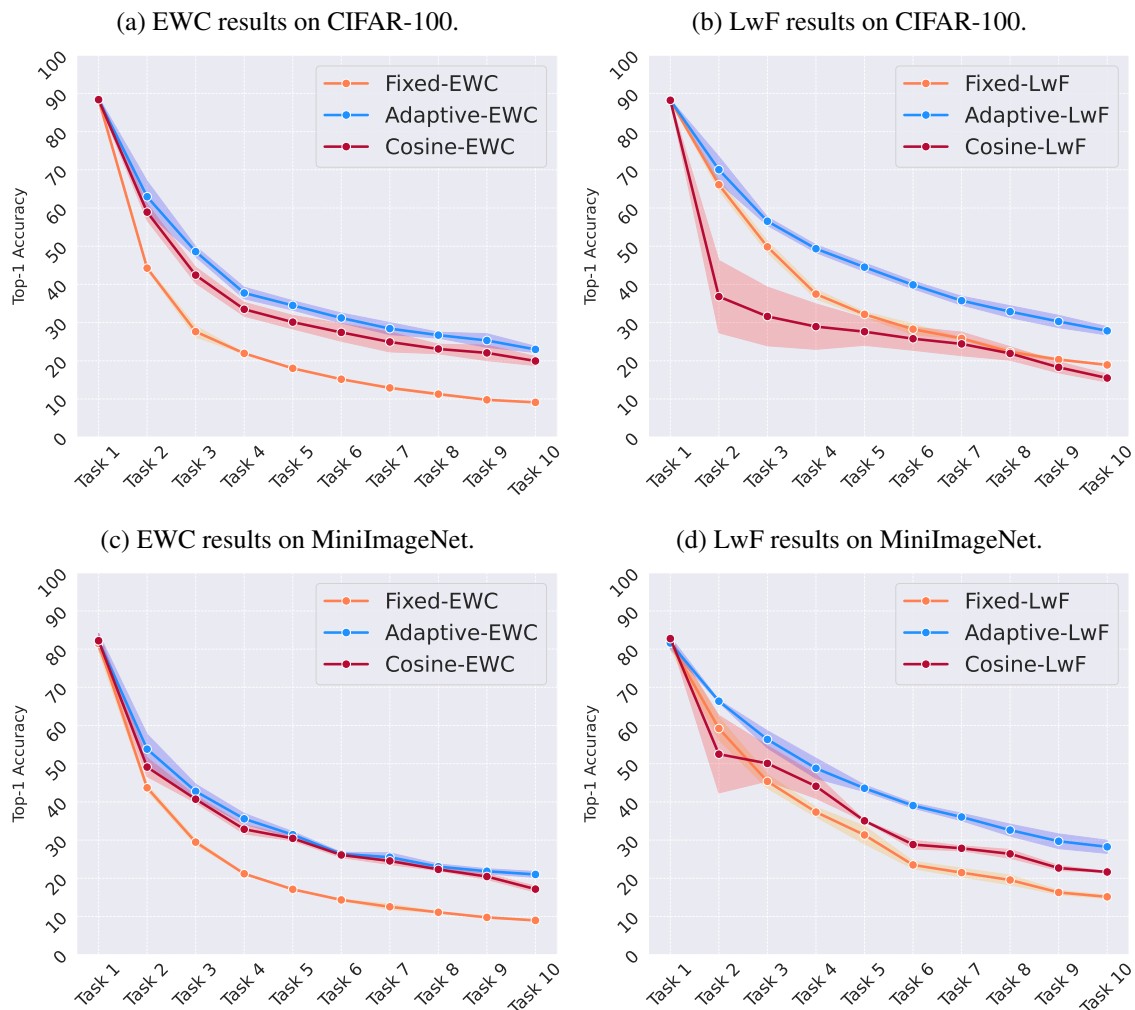

Figure 2: Comparing fixed *vs.* adaptive regularization on CIFAR-100 and MiniImageNet. Regularizing adaptively outperforms fixed regularization by a large margin, across all incremental learning steps.

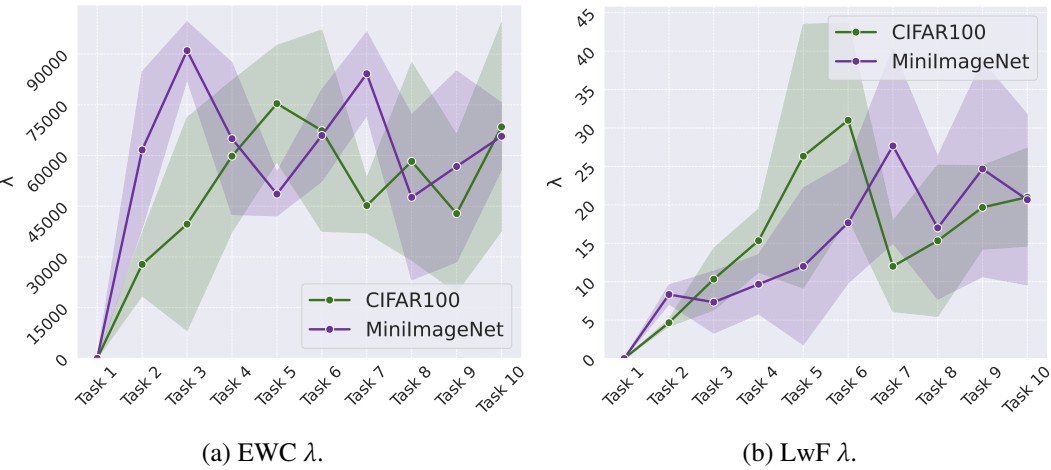

(a) EWC $\lambda$.

(b) LwF $\lambda$.

Figure 3: $\lambda$ values with respect to incremental learning tasks. The model learns to adjust the magnitude of regularization per-learning task with high variability. Task 1 always starts with 0 as there is no regularization at the initial task.

As is visible, our method is able to adjust the regularization magnitude according to the learning task with high variance. This claims that our model with a given objective function was able to identify task difficulty (dissimilarity) by itself. When the new task is not similar to the old tasks, the model reduces the regularization strength at a certain point so that it does not forget the old tasks but learns the new task as well. In contrast, if the new task and the old tasks are similar, a higher regularization strength is selected by the model, preventing the loss of previous knowledge. It reaffirms that the fixed regularization scheme is unnatural, and explains the superior performance reported in previous sections.

## 6 Limitations and Broader Impact

Our work has three limitations that we want to acknowledge:

- **Storage:** Our approach requires a subset of the validation set from previous classes. Although this is similar to replay-based techniques, our objective is not to introduce a new incremental learning technique. Instead, we demonstrate the feasibility of learning to regularize for regularization-based incremental learning. Stored exemplars are only used for tuning the regularization coefficient $\lambda$.

- **Efficiency:** Our method requires auto-tuning at each incremental learning step to identify the optimal regularization magnitude. This process scales linearly with the number of tasks, which adds a minor but not insignificant training cost. In a broader context, as we move towards longer, larger-scale incremental learning scenarios, our method is expected to improve the learning efficiency by automating the selection of $\lambda$, as typically researchers perform multifold cross validation to find optimal parameters per-task via grid search, eventually reducing the required carbon footprint.

- **Baselines:** In this paper, we experimented with two complementary baselines, EWC and LwF, to assess the generality of our approach for prior-based and distillation-based techniques. Extending this work would involve experimenting with additional techniques to see how adapting to the current task affects performance. Fortunately, our proposal is straightforward and method-agnostic, so it can be easily applied to other baselines with no algorithmic modification.

## 7 Conclusion

This work addresses class-incremental learning, where learning tasks arrive as a sequence of mutually exclusive categories. We propose a proof-of-concept study to test whether adapting to the current task is necessary in class-incremental learning. We enhance two popular regularization approaches, LwF (Li and Hoiem, 2017) and EWC (Kirkpatrick et al., 2017), by enabling the adjustment of regularization weight per-learning task. We auto-tuned the regularization weight using the black-box Bayesian Optimization technique of multivariate tree parzen estimator.

Through experiments on CIFAR-100 and MiniImageNet, we demonstrate that tuning regularization per-learning task is critical and leads to significant improvements in accuracy and transfer, regardless of the technique used. This suggests that the constant regularization assumption is a fundamental limitation of regularization-based approaches. We hope our work will inspire future research for further auto-tuning the otherwise hand-crafted parameters in incremental learning.

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

## 9 Appendix

We have built our experiments on PyCIL (Zhou et al., 2021a) (MIT License) and RayTune (Liaw et al., 2018) (Apache License 2.0) frameworks. Our experiments were run on NVIDIA A100 taking around 148 GPU hours. Total emissions are estimated to be around 15 kg $CO_2$.

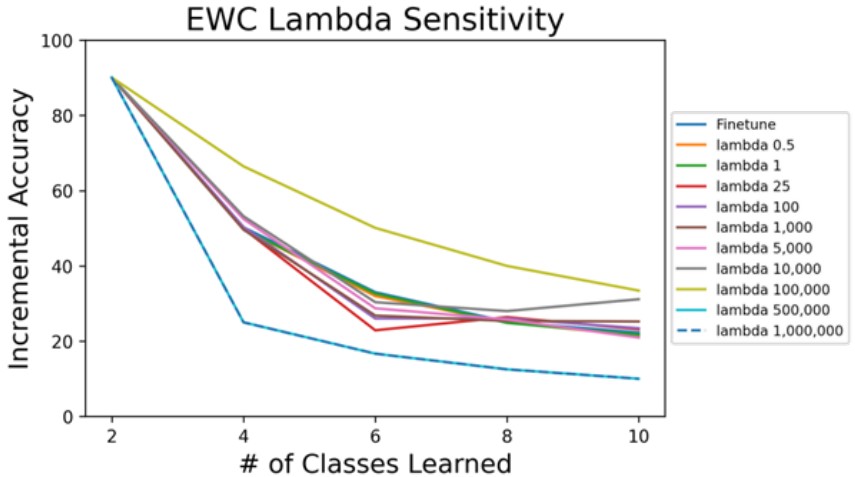

Figure 4: EWC sensitivity analysis on CIFAR-10.

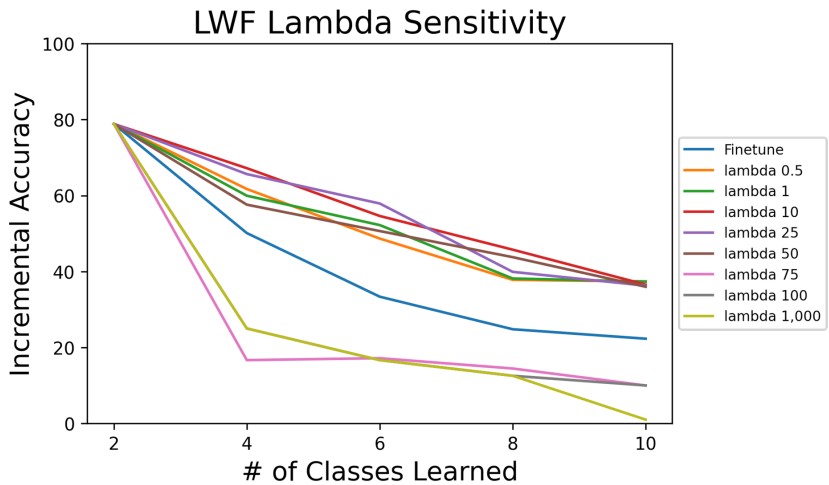

Figure 5: LwF sensitivity analysis on CIFAR-10.

