# OpenReview forum: "Adaptive Regularization for Class-Incremental Learning"
_automl.cc/AutoML/2023/Conference — AutoML 2023 Workshop_

### Official Review · Reviewer_6HjX · 2023-04-09

**Potential Impact On The Field Of Automl Rating:** 2
**Technical Quality And Correctness Rating:** 2
**Clarity:** The paper is mostly well written and …
**Clarity Rating:** 4

**Summary Of Contributions:**

The paper proposes to change the strength of a regularizer on-the-fly for class-incremental learning. To this end, a novel BO-style approach is introduced which changes the $\lambda$ hyperparameter in-between class samples.

**Actions Required To Increase Overall Recommendation:**

I am willing to increase my score when
* all the missing details of how the hyperparameters are set are given,
* how the hyperparameters were tuned,
* an explanation is given for why different setups for the different methods are needed,
* a discussion of related work on dynamic adaptation is included,
* at least one stronger, dynamic baseline is considered,
* answers to the research questions provide more insights

**Overall Review:**

I find the work interesting and would like to see more work in this direction. However, I am left with the feeling that the work is not yet finished. Too many details are still missing to make it clear how everything comes together, to make it a concise paper. The experiments seem fairly limited, though promising enough to keep working in the direction. In particular, in its current form the baselines seem to be not particularly meaningful and answers to research questions not insightful. Thus, my overall vote is for rejection.

**Potential Impact On The Field Of Automl:**

The impact seems somewhat limited. In AutoML, it is well known that (some) hyperparameters need to be dynamically adapted. There have been various methods proposed to do so, ranging from tuning the parameters of a schedule to full on learning of novel schedules. Still, to the best of my knowledge, it has not before been shown that regularization (strength) is best adapted while learning. While the "class-incremental learning" setting might be a special case where this is needed, it still provides evidence for need of dynamic tuning in that regard, which could inspire future work in the AutoML community.

**Reproducibility (Optional):**

As pointed out in my assessment of "technical quality and correctness", I fear that the work, in its current state, is not reproducible. Too many details seem to be missing.

**Review Confidence:**

4: You are confident in your assessment, but not absolutely certain. It is unlikely, but not impossible, that you did not understand some parts of the submission or that you are unfamiliar with some pieces of related work.

**Review Rating:**

3: Reject: For instance, a paper with technical flaws, weak impact, and/or weak evaluation.

**Review Summary:**

While the work presents an interesting problem and solution approach, it seems not yet ready for publication.

**Technical Quality And Correctness:**

### Experimental setup & Hyper-hyperparameters
In line 144 it is stated that "[t]he estimator is trained on the validation set accumulated across incremental learning sessions." Why is it necessary for the data to be from the validation set here? What would change if, instead of keeping 20% of the validation data of each task, keeping 4% of the training data of each task? There is no justification given as to why the adaptation of $\lambda$ is learned with respect to the validation, rather than the training data.

When the experiments are presented, only training and test-set size are given but not how large the validation set(s) are.

It is also not clear how the hyper-hyperparameters have been determined. Why is it necessary to have such a drastically different setup for the use-case with EWC to that with LwF? While in the Checklist, it is repeatedly stated that it is stated how the hyperparameters of their method have been determined (often referring to the appendix) I could not find this information. As such, it seems that the proposed method is very dependent on expert knowledge to properly setup the adaptation mechanism, which should have been stated as a limitation. Further, with all these missing details, I doubt that the experiments are fully reproducible, even if the code is provided. From the paper itself, it is not clear how equation 2 is actually optimized. To me it would have been much nicer to include pseudocode of how TPE is used to optimize equation 2 to give more insights.

Experiments are reported with only three seeds, which seems far too low. While I appreciate that experiments are costly, in the Checklist it is stated that the experiments are run on surrogate benchmarks, which should have made costs close to zero. However, even if the benchmarks used are not actually surrogates (as they are not from my understanding) than a few more seeds would need to be evaluated for better understanding of the performance of the method.

### Missing constraint
For equation 2, the way everything is stated, $\lambda$ should trivially be set to $-\infty$ to optimize equation 2.  A constraint on $\lambda \geq 0$ should fix this issue though.

### Choice of baselines
It is well known in the AutoML community (and various other AI fields) that some hyperparameters are best dynamically adapted. Thus, a comparison only between static baselines to the proposed dynamic baseline seems not enough. At least a hand-crafted schedule seems to be an appropriate baseline. Further, it is not clear how the static $\lambda$ was determined for the baselines. If there was some statement or evidence that the static baseline against which is compared is optimized (or even optimal) then the comparison would be somewhat meaningful. However, without that it becomes a comparison to any static baseline that could work arbitrarily bad.

### Missing insights from the experiments
While the experiments aim to shed light on three given research questions, there are often not a lot of insights to be gained.
For example RQ3 asks "How does the predicted $\lambda$ values differ across incremental learning steps?". To answer this question, two plots are given that show $\lambda$ changing drastically. However, there are no insights given if that is actually sensible. In section three, two scenarios are introduced which the authors say are the reason for need of adapting $\lambda$. This could (and arguably should) be taken up again when answering RQ3 to show that, whenever a similar task is introduced a small $\lambda$ can be used and for a very dissimilar task a large lambda is needed. Otherwise, the reader is left wondering what the plots are supposed to show, since there is no clear trend across datasets visible nor similar behavior for the same backbone method.

### No discussion of related work on dynamic adaptation
A big shortcoming of the work is that there is no discussion of related work on dynamic hyperparameter optimization. Such a discussion would have potentially showed advantages of the proposed method. For example, such a discussion could have pointed out that population-based training, while capable of dynamic adaptation, might be far too costly for the considered setting. Other approaches, such as dynamic algorithm configuration, might have a cheaper online adaptation step but suffer from vastly more expensive offline training phase. Hand-crafted schedules, while potentially high performing, are very difficult to create with frequent, drastic changes.

### Minor details:
* It is not clear why $\lambda$ is constant for the first two tasks (as shown in Figure 3)
* In line 174 it should be "To what extend **does** adaptive ..."
* Some of the citations do not seem correct. E.g.
  * Deng et al., 2009 is given as reference to MiniImageNet but the reference is actually to ImageNet. AFAIK, the correct reference is Vinyals et al., 2016 "Matching Networks for One Shot Learning"
  * Turner et al., 2021 is given as reference for Bayesian Optimization. A more appropriate reference would be Snoek et al., 2012 "Practical Bayesian Optimization of Machine Learning Algorithms".
   * Similarly, references for bi-level optimization and gradient-based optimization seem not the most appropriate

---

> ### Author Response · Authors · 2023-05-01
> **Respond to Reviewer 6HjX**
>
> We thank the reviewer for their detailed review. We answer the questions below.
>
> Comment: In line 144 it is stated that "[t]he estimator is trained on the validation set accumulated across incremental learning sessions." Why is it necessary for the data to be from the validation set here? What would change if, instead of keeping 20% of the validation data of each task, keeping 4% of the training data of each task? There is no justification given as to why the adaptation of is learned with respect to the validation, rather than the training data. When the experiments are presented, only training and test-set size are given but not how large the validation set(s) are.
>
> Reply: We build our validation set with 20% of training data after each incremental step. For example, if we are training on Task 3, our validation set consists of 20% of Task1 + 20% of Task2 + 20% of Task3 to select the best regularization strength. The training data only includes current task data whereas validation data includes previous task + current task.
>
> Comment: It is also not clear how the hyper-hyperparameters have been determined. Why is it necessary to have such a drastically different setup for the use-case with EWC to that with LwF? While in the Checklist, it is repeatedly stated that it is stated how the hyperparameters of their method have been determined (often referring to the appendix) I could not find this information. As such, it seems that the proposed method is very dependent on expert knowledge to properly setup the adaptation mechanism, which should have been stated as a limitation.
>
> Reply: To determine the optimal range of the regularization hyperparameter, we conducted a sensitivity analysis using the CIFAR 10 dataset. Our findings indicate that the lambda hyperparameter performs optimally within the range of [1, 100000] for Elastic Weight Consolidation (EWC) and [1, 50] for Learning without Forgetting (LwF). Therefore, we adopted the respective ranges in further experiments. We added CIFAR 10 experiment results to Appendix. For the rest of the hyperparameters, we used the default hyperparameters of the PYCIL library.
>
> Comment: For equation 2, the way everything is stated, should trivially be set to optimize equation 2. A constraint on should fix this issue though.
>
> Reply: Thank you. We updated our equation.
>
> Comment: It is well known in the AutoML community (and various other AI fields) that some hyperparameters are best dynamically adapted. Thus, a comparison only between static baselines to the proposed dynamic baseline seems not enough. At least a hand-crafted schedule seems to be an appropriate baseline. Further, it is not clear how the static was determined for the baselines. If there was some statement or evidence that the static baseline against which is compared is optimized (or even optimal) then the comparison would be somewhat meaningful. However, without that it becomes a comparison to any static baseline that could work arbitrarily bad.
>
> Reply: We conduct more experiments with the reviewer’s suggestion. We compared our adaptive approach with a very strong baseline: cosine annealing. We updated the results of our paper.
>
> Comment: While the experiments aim to shed light on three given research questions, there are often not a lot of insights to be gained. For example RQ3 asks "How does the predicted values differ across incremental learning steps?". To answer this question, two plots are given that show changing drastically. However, there are no insights given if that is actually sensible. In section three, two scenarios are introduced which the authors say are the reason for need of adapting. This could (and arguably should) be taken up again when answering RQ3 to show that, whenever a similar task is introduced a small can be used and for a very dissimilar task a large lambda is needed. Otherwise, the reader is left wondering what the plots are supposed to show, since there is no clear trend across datasets visible nor similar behavior for the same backbone method.
>
> Reply: We design our objective function to let the model discover the task difficulty itself and select the best lambda accordingly. Different seeds yield different task orderings and as can be seen from the confidence interval in the Figure 3 model changes the lambda value drastically for different task orderings.
>
> Comment: It is not clear why is constant for the first two tasks (as shown in Figure 3)
>
> Reply: We changed Figure 3 to prevent confusion and made the explanation under the figure.
>
> Comment: Some of the citations do not seem correct.
>
> Reply: Thank you. We updated our references with your suggestion.

---

> > ### Comment · Reviewer_6HjX · 2023-05-02
> > **Reply to rebuttal**
> >
> > Thank you for addressing some of my concerns.
> >
> > **Regarding tuning $\lambda$ with respect to the validation set:** I understand how you create the dataset for adapting the regularization strength. Your explanation reads as the choice of using validation data was simply done out of convenience not out of necessity. Hence, I'm still asking myself if it is important in this setting to tune with respect to held-out data, i.e. data that was not used to train the network weights. Do you have any insights on this?
> >
> > **Regarding hyper-hyperparameter tuning of the presented method:** I don't understand how the grid-search you present in the appendix informed the choice of configuration space. Firstly, it is not explained in the plots what "Finetune" represents. Secondly, while I can somewhat see how you derived the configuration space for LWF, for EWC it looks like only "100,000" and maybe "10,000" (based on the performance increase between 8&10 #of classes learned) would build a reasonable configuration space. Further, judging from these plots, the static baseline with a fixed $\lambda$ of 1 does not make sense. In the LWF case the best static value seems to be 10 and for EWC 100,000. Thus, those should serve as baselines.
> >
> > **Regarding the additional *strong baseline*:** The plots in the appendix make me seriously doubt that the new cosine baseline is sensible. Since it is stated that this baseline starts from the largest value in the range (100,000 for EWC and 50 for LWF) and is decreased to 1 following a cosine decay, this baseline does not seem sensible. In the EWC case the plots in the appendix show that larger values are to be preferable and in the LWF case smaller ones. Thus I don't see how a smooth change between these would make sense. A strong baseline would not be time-dependent here but would exploit knowledge about the tasks. Especially since the experiments consider different seeds which give you different task orderings. While I appreciate the effort of including a dynamic baseline, this does not seem to be a sensible choice.
> >
> > **Regarding the objective function:** How exactly does the objective function cover task difficulty? This seems to be a rather vague notion of difficulty and not based in theoretical understanding of the objective. Due to the changing task orders, the x-axis label of Figure 3 is wrong. It should rather be the same as in the plots in the appendix. However, this plot looses a lot of interpretability by averaging over different task orderings over a small set of seeds. Given the law of large numbers, when running many more seeds, we would start to maybe see some interesting average behavior. This plot would be much more informative if it was split into 3 plots, for each individual seed, and included an analysis of task similarity in the ordering.
> >
> > **I have also read all other reviews and rebuttals and, as of yet, am *not* convinced to increase my score.**

---

### Official Review · Reviewer_fLq8 · 2023-04-11

**Potential Impact On The Field Of Automl Rating:** 2
**Technical Quality And Correctness Rating:** 3
**Clarity Rating:** 3

**Summary Of Contributions:**

This paper proposes to adaptively change the regularization coefficient for training deep NNs in the class-incremental setting. The adaptive regularization coefficient is optimized using an efficient Bayesian optimizer (TPE), w.r.t. the penalized/regularized loss function computed over a subset of the data points of the current and past tasks. Two regularizers, LwF and EWC were employed in the experiment, where the author took the CIFAR-100 and MiniImageNet data sets and ResNet-32 backbone. The results clearly showed that with the adaptive scheme, the performance metrics of the model are significantly improved on those data sets, and indeed, the optimal choice of the regularization coefficient differs from task to task quite a bit.

**Actions Required To Increase Overall Recommendation:**

Some steps to take to improve the paper:

- Please explain how two ranges [1, 1e5] and [1, 50] are determined for EWC and LwF, respectively.

- Please study the sensitivity of the lambda parameter in each new task in terms of (1) the performance of learning the new task and (2) the effect of catastrophic forgetting.

- Potentially, please provide an explanation/analysis of why, for some tasks, the optimal choice of lambda is larger/smaller than other tasks.

**Clarity:**

Most of my concerns are exactly the same in the "Technical Quality And Correctness" section above, which are:

1. Around lines 113 - 115, you formulate the adaptive lambda as a time series whose current value is conditioned on the past. However, I did not see immediately how such a consideration is realized in your HPO/optimization approach. When seeing this formulation, I thought you like to build a GPR model to directly predict the new lambda setting based on features of the data set, which is a bit confusing to me.

2. The viable range of the regularization coefficient ofc depends on the range of the loss values and the chosen regularization term. Could you explain how two ranges [1, 1e5] and [1, 50] are determined for EWC and LwF, respectively?

3. What is the fixed lambda used for the control group in your experiment? I did not find it in the paper.

4. In Figure 3, I did not see the variability of the lambda reported across three independent runs. This is important to understand if the evolution of lambda values is reliable.

5. In the motivation, you mention that the optimal choice of lambda is coupled with the "complexity" of the tasks. I would like to see a deeper discussion on this when showing the evolution of the optimal lambda values.

**Overall Review:**

Pros:

The paper has a strong motivation to study and improve the regularization of deep NN models, where the proposed HPO-based method and the experimentation look solid and correct. Also, the author nicely formulated three research questions and answered them with numerical evidence. The limitation of the current work is also addressed.

Cons:

To me, the proposed approach does not touch the intrinsic learning mechanism in the class-incremental scenario. The reader does not gain much insight from the numerical study, particularly on the interpretation of the optimal lambda value.

**Potential Impact On The Field Of Automl:**

This paper brings an interesting application of the SOTA AutoML/HPO techniques to class-incremental learning. Also, it addresses the necessity of evolving/adapting the regularization coefficient in deep learning. It certainly opens quite a few intriguing research questions, e.g., what is the relation between the optimal regularization coefficient with the similarity between classification tasks and the difficulty thereof?

**Review Confidence:**

4: You are confident in your assessment, but not absolutely certain. It is unlikely, but not impossible, that you did not understand some parts of the submission or that you are unfamiliar with some pieces of related work.

**Review Rating:**

6: Borderline Leaning Accept: Technically sound paper where reasons to accept outweigh reasons to reject. Please use sparingly.

**Review Summary:**

Based on my comments above, I think the paper is quite standard in terms of proposing new learning techniques. However, it lacks a deeper explanation/investigation of the impact/relation between the choice of the regularization coefficient and the "similarity" of different tasks. Thereby, I recommend a borderline acceptance of it.

**Technical Quality And Correctness:**

The paper is generally well-written and easy to follow. There are quite a few things to improve and clarify.

As for the motivation of the work:

1. In the intro. (lines 39 and 40), ".. treating the regularization magnitude as a latent variable that should be adjusted based on the current state of the learner and the complexity of the task.." -> what do you mean exactly by the complexity of the task? Sample complexity or something else?

2. I found that Figure 1 is not really needed. The idea it conveys is clear from the text.

3. In section 3, regarding "Low Plasticity and High Stability/High Plasticity and Low Stability," I wonder if the intuitive argument is valid. Is it always true that incrementally learning a new class car is harder than learning a bird? Is there any empirical evidence? To me, it should be the case that the conditional distribution/classification function of the new class is quite far from the learned ones in some function/model spaces.

Some technical aspects:

1. Around lines 113 - 115, you formulate the adaptive lambda as a time series whose current value is conditioned on the past. However, I did not see immediately how such a consideration is realized in your HPO/optimization approach. When seeing this formulation, I thought you like to build a GPR model to directly predict the new lambda setting based on features of the data set, which is a bit confusing to me.

2. The viable range of the regularization coefficient ofc depends on the range of the loss values and the chosen regularization term. Could you explain how two ranges [1, 1e5] and [1, 50] are determined for EWC and LwF, respectively?

3. What is the fixed lambda used for the control group in your experiment? I did not find it in the paper.

4. In Figure 3, I did not see the variability of the lambda reported across three independent runs. This is important to understand if the evolution of lambda values is reliable.

5. In the motivation, you mention that the optimal choice of lambda is coupled with the "complexity" of the tasks. I would like to see a deeper discussion on this when showing the evolution of the optimal lambda values.

---

> ### Author Response · Authors · 2023-05-01
> **Respond to Reviewer fLq8**
>
> We thank the reviewer for their detailed review. We answer the questions below.
>
> Comment: In the intro. (lines 39 and 40), ".. treating the regularization magnitude as a latent variable that should be adjusted based on the current state of the learner and the complexity of the task.." -> what do you mean exactly by the complexity of the task? Sample complexity or something else?
>
> Reply: Here the task complexity refers to the difference between the current task and previous tasks. In our study, we selected task sequences in a completely random manner without any consideration for their difficulty levels. Instead, we relied on the model's ability to identify the degree of difficulty on its own by evaluating the loss function. For instance, if a new task is perceived to be more difficult (i.e. less similar) compared to previous ones, the regularization strength is decreased to encourage more learning/adaptation. Conversely, if the new task was similar or less difficult, the regularization strength was kept high to prevent knowledge degradation.
>
> Comment: In section 3, regarding "Low Plasticity and High Stability/High Plasticity and Low Stability," I wonder if the intuitive argument is valid. Is it always true that incrementally learning a new class car is harder than learning a bird? Is there any empirical evidence? To me, it should be the case that the conditional distribution/classification function of the new class is quite far from the learned ones in some function/model spaces.
>
> Reply:  In the related section, we are also supporting the same idea. If the distribution of previous tasks are different than the task at hand, learning the new task would be harder, and lower regularization strength is required. On the other hand, if the distributions are similar then it would be easier to learn the new task. We do not claim that “learning car is always harder than learning the bird”, it depends on the previous condition. We clarify and give related information in Section 3, Low Plasticity and High Stability vs High Plasticity and Low Stability part.
>
> Comment: Around lines 113 - 115, you formulate the adaptive lambda as a time series whose current value is conditioned on the past. However, I did not see immediately how such a consideration is realized in your HPO/optimization approach. When seeing this formulation, I thought you like to build a GPR model to directly predict the new lambda setting based on features of the data set, which is a bit confusing to me.
>
> Reply: In the formulation, we consider the time (number of tasks in our case) by optimizing towards the validation set which includes both current and previously learned tasks. In formulation, we refer to it as Vt.
>
> Comment: The viable range of the regularization coefficient ofc depends on the range of the loss values and the chosen regularization term. Could you explain how two ranges [1, 1e5] and [1, 50] are determined for EWC and LwF, respectively?
>
> Reply: To determine the optimal range of the regularization hyperparameter, we conducted a sensitivity analysis using the CIFAR 10 dataset. Our findings indicate that the lambda hyperparameter performs optimally within the range of [1, 100000] for Elastic Weight Consolidation (EWC) and [1, 50] for Learning without Forgetting (LwF). Therefore, we adopted the respective ranges in further experiments. We added CIFAR 10 experiment results to Appendix.
>
> Comment: What is the fixed lambda used for the control group in your experiment? I did not find it in the paper.
>
> Reply: Thank you. We added this information in Section 3.2 where we mention the ranges of EWC and LwF.
>
> Comment: In Figure 3, I did not see the variability of the lambda reported across three independent runs. This is important to understand if the evolution of lambda values is reliable.
>
> Reply: We change our Figure 3 plot, we averaged and plotted the confidence intervals of the selected regularization strength values for all 3 different seeds.
>
> Comment: In the motivation, you mention that the optimal choice of lambda is coupled with the "complexity" of the tasks. I would like to see a deeper discussion on this when showing the evolution of the optimal lambda values.
>
> Reply: In the Experimental Results section we provided more discussion regarding to your suggestion and colored it.

---

> > ### Comment · Reviewer_fLq8 · 2023-05-02
> > **Reply to rebuttal**
> >
> > Thank you for clarifying my concerns.
> >
> > *Re. the relation between the task complexity and the regularization parameter* I fully understand your intuition and approach. But, what I need to see is a ground truth - how difficult to solve each task in the sequence independently? I am not so sure if there exists such a measure. Or, like you wrote, a similarity measure among the tasks, which will help justify you intuition/motivation, that is "when a new task is perceived to be more difficult (i.e. less similar) compared to previous ones, the regularization strength is decreased to encourage more learning/adaptation"
> >
> > *Re. adaptive lambda as a time series* I understand your approach - there is an implicit dependency on the past task. But, I wouldn't call it a time-series modeling (maybe I got it wrongly); It is just an adaptive approach.

---

> > > ### Author Response · Authors · 2023-05-08
> > > **Respond to Reviewer fLq8**
> > >
> > > **Comment:** adaptive lambda as a time series I understand your approach - there is an implicit dependency on the past task. But, I wouldn't call it a time-series modeling (maybe I got it wrongly); It is just an adaptive approach
> > >
> > > **Reply:** Thank you for your constructive comment. Actually, we agree that it is not modeled as a time series, but rather as an incremental series conditioned on the current learning task and all previous tasks. We reframed the related content in our paper.

---

### Review · Reproducibility_Reviewer_AfRG · 2023-04-11

**Completeness Of Code And Dataset Supplement Rating:** 2
**Usability And Ease Of Reproducibility Rating:** 2

**Actions Required To Increase The Reproducibility And Overall Recommendation:**

The authors should include code for reproducing all figures and tables in the paper. Furthermore, the authors should include code to parse the results.

The file "rmm_train.py" cannot be run without running into errors. While this file is not referred to in the README, this is confusing.

Overall, it would be good to have a file that runs all experiments and creates the figures (while having command line switches to control the number of repetitions, etc.)



**Completeness Of Code And Dataset Supplement:**

The code to run the method with the two regularization methods is provided. Unfortunately, the code for reproducing figures or tables seems to be missing. This certainly could be fixed for the rebuttal, but so far, I cannot reproduce the results without major efforts as it is unclear how to parse them.

**Overall Reproducibility Review:**

The paper proposes a method for improving regularization in transfer learning using adaptive regularization and Bayesian optimization. The authors provide code to solve the problem for two types of regularization, but the code for reproducing figures or tables is missing. The reproducibility checklist appears incomplete, and important aspects are missing to reproduce the results. As a result, the figures cannot be reproduced, and the results cannot be parsed without major efforts. Thus, the overall reproducibility of the paper appears to be low, and additional information or clarification is required to reproduce the results.

**Review Confidence:**

3: You are fairly confident in your assessment. It is possible that you did not understand some parts of the submission or that you are unfamiliar with some pieces of the code or data.

**Review Rating:**

4: Weak Reject, you were not able to reproduce some critical aspects of the paper, but believe it is likely possible with additional effort.

**Review Summary:**

The paper proposes a method for improving regularization in transfer learning using adaptive regularization and Bayesian optimization. However, the paper lacks reproducibility as the code for reproducing figures or tables is missing, and important aspects are missing to reproduce the results. All issues could be addressed during the rebuttal. In its current state, the paper should not be accepted due to reproducibility concerns.

While the paper proposes an interesting method for improving regularization in transfer learning, the lack of reproducibility limits its usefulness to the community. Reproducibility is a critical aspect of scientific research, and without providing the necessary resources to reproduce the results, the paper's potential impact is significantly reduced.

**Summary Of Necessary Code And Dataset Supplement:**

This paper uses adaptive regularization to improve the regularization used in transfer learning. The authors argue that adaptive regularization essentially boils down to solving an expensive black-box optimization problem for which they use Bayesian optimization. This generic

The authors provide the code to solve this problem for two types of regularization.

**Usability And Ease Of Reproducibility:**

As discussed above, important aspects are missing to reproduce results. As of now, results are not reproducible without major efforts. The figures cannot be reproduced, and the results cannot be parsed.

---

> ### Author Response · Authors · 2023-05-01
> **Respond to Reproducibility Reviewer AfRG**
>
> We thank the reviewer for their detailed review. We answer the questions below.
>
> Comment: Unfortunately, the code for reproducing figures or tables seems to be missing. This certainly could be fixed for the rebuttal, but so far, I cannot reproduce the results without major efforts as it is unclear how to parse them.
>
> Comment: The file "rmm_train.py" cannot be run without running into errors. While this file is not referred to in the README, this is confusing.
>
> Reply: Thank you so much for pointing this out. As you requested, we added the code for all our figures. They are plotted based on the logs after completing the experiments. After running the referred files in the README you will find the detailed results under the ‘logs’ folder. Based on your comments, we updated our README file to help readers reproduce out results. There was also a tiny mistake in num_samples which is now fixed. Please refer to the new README file and please run “trainer_adaptive_lwf.py” and “trainer_adaptive_ewc.py” to reproduce the results.

---

### Official Review · Reviewer_Pz2q · 2023-04-12

**Potential Impact On The Field Of Automl Rating:** 2
**Technical Quality And Correctness:** I don't see any issues related to this.
**Technical Quality And Correctness Rating:** 3
**Clarity Rating:** 3

**Summary Of Contributions:**

To summarize, the research investigates the importance of adaptive regularization in Class-Incremental Learning, which dynamically adjusts the regularization strength based on the complexity of the current task. The study focuses on class-incremental learning, where learning tasks arrive as a sequence of distinct categories, and proposes a proof-of-concept study to test the necessity of adapting to the current task. The authors enhance two popular regularization approaches, LwF and EWC, by allowing the adjustment of regularization weight per learning task. The regularization weight is auto-tuned using the multivariate tree parzen estimator, a black-box Bayesian Optimization technique. The study demonstrates that tuning regularization per-learning task is crucial and leads to significant improvements in accuracy and transfer, regardless of the technique used. These results suggest that the assumption of constant regularization is a fundamental limitation of regularization-based approaches. The authors hope that their work will encourage future research on further auto-tuning the manually defined parameters in incremental learning.

**Actions Required To Increase Overall Recommendation:**

Including experiments that explore the variation of the regularization factor value (such as gradient) in response to differences in difficulty between the current and previous tasks would make the study more compelling. To be more precise, it would be beneficial to include the adaptive regularization factor values based on the difficulty of the task in Figure 3. This would provide greater insight into how the lambda value changes in relation to task difficulty, rather than only displaying the lambda value for each task. I recommend conducting an ablation study on various methods of Bayesian Optimization.





**Clarity:**

This paper is written in a clear and readable manner for readers to easily understand the concept of class-incremental learning. However, the explanation of BO is somewhat insufficient. Specifically, I hope that there will be a more detailed explanation of how BO was utilized in section 3.2.

**Overall Review:**

I fully understand the problems with the existing class-incremental learning process and why this is necessary. The results of the study with and without this method are also very intriguing. Furthermore, the writing is clear and easy to follow for readers, especially with the well-illustrated big picture of the class-incremental learning explanation. However, the experiment scale is limited to proof-of-concept, and there are not enough experiments. It would be more interesting to add experiments that investigate how the regularization factor value changes (e.g., gradient) depending on the difference in difficulty between the current task and the previous task. It would also be helpful to show the results of applying the method to other approaches, not just EWC and LwF.

**Potential Impact On The Field Of Automl:**

While adopting the BO methodology to automate adaptation in class-incremental learning is admirable, it is a rather straightforward approach. It would be beneficial to have a more in-depth study on search strategies.

**Review Confidence:**

4: You are confident in your assessment, but not absolutely certain. It is unlikely, but not impossible, that you did not understand some parts of the submission or that you are unfamiliar with some pieces of related work.

**Review Rating:**

5: Borderline Leaning Reject: Technically sound paper where reasons to reject nonetheless outweigh reasons to accept. Please use sparingly.

**Review Summary:**

You provided clear reasons for why the proposed method is necessary, how it could be applied to existing methods, and the benefits it could bring. The writing is also clear and easy to understand. However, as previously mentioned, it is disappointing that there is a lack of experiments and the scale of the study is small.

---

> ### Author Response · Authors · 2023-05-01
> **Respond to Reviewer Pz2q**
>
> We thank the reviewer for their detailed review. We answer the questions below.
>
> Comment: It would be beneficial to have a more in-depth study on search strategies.
>
> Reply: Our early analysis of these regularization parameters showed that the response surface is smooth, but also quite complex - it is very hard to predict this surface for any given current (source) and new (target) task. Although we didn’t directly evaluate derivative-free algorithms, we expect them to be quite suboptimal given the complexity of the response surface. Bayesian Optimization is a probabilistic approach that efficiently explores the most promising regions of the hyperparameter space, and can therefore adapt very efficiently to the response of the model when encountering a new task. It has been shown to work very well when the objective function is smooth and continuous, and the hyperparameters are few and continuous or categorical. On the other hand, genetic search is much less efficient in this particular setting, e.g. it would have to go through many crossover and random mutation operators across many generations. This is better suited for high-dimensional search spaces, but we expected it to be much less efficient in this particular setting. Since we are optimizing towards a single hyperparameter (regularization strength) we selected Bayesian optimization in our study, and it works well. Another reason we selected Bayesian optimization is that it opens the door to a range of meta-learning approaches (e.g. warm-starting and surrogate model transfer) which could be explored in future work to learn across tasks and adapt models even faster.
>
> Comment: explanation of BO is somewhat insufficient. Specifically, I hope that there will be a more detailed explanation of how BO was utilized in section 3.2.
>
> Reply: Thank you for bringing this up. We extended the paper and added how BO is utilized precisely in the paper and color-coded the additions in Section 3.2.
>
> Comment: It would be more interesting to add experiments that investigate how the regularization factor value changes (e.g., gradient) depending on the difference in difficulty between the current task and the previous task. It would also be helpful to show the results of applying the method to other approaches, not just EWC and LwF.
>
> Comment: To be more precise, it would be beneficial to include the adaptive regularization factor values based on the difficulty of the task in Figure 3. This would provide greater insight into how the lambda value changes in relation to task difficulty, rather than only displaying the lambda value for each task.
>
> Reply for both comments: These are certainly very interesting suggestion that we would love to explore in future work. For this particular study, the task sequences were chosen completely random, and no ordering was made according to the difficulty of the tasks. We did this on purpose to show that adaptive regularization works regardless of the difficulty of the new task or the differences in difficulties between tasks. On the contrary, we let the model find the degree of difficulty itself based on the loss function. For example, if the new task is different (difficult) from the previous tasks, the regularization strength is lowered, allowing the new task to be learned. In the opposite case, knowledge degradation is minimized by keeping the regularization strength high. Regarding this, we added a more clear explanation after Figure 3. That said, it would certainly be interesting to explore this further and learn how to include task difficulty in future versions of this approach. Elastic Weight Consolidation and Learning without Forgetting are established and widely adopted techniques in the field of continual learning, serving as strong baselines against which newly proposed methods are often compared. Despite recent research benefiting from combining regularization and memory-based approaches, the focus of our study was to demonstrate the effectiveness of adaptive regularization in continual learning, with particular emphasis on well-established pure regularization methods. As stated in our limitation and conclusion sections, this research may serve as inspiration for future work exploring further refinement of hand-crafted parameters in incremental learning through auto-tuning.

---

### Official Review · Reviewer_wAxZ · 2023-04-12

**Potential Impact On The Field Of Automl Rating:** 2
**Technical Quality And Correctness Rating:** 2
**Clarity Rating:** 4

**Summary Of Contributions:**

In this paper, the authors attempt to investigate the necessity of adaptive regularization in the class-incremental setting. To this end, the authors use a multivariate tree-structured parzen estimator (TPE) from Bayesian optimization to predict the best regularization strength per task. The analysis is done with two baselines in the class-incremental setting EWC and LwF, representing prior-based and distillation-based methods, and focuses on image classification on Split-CIFAR100 and Split-MiniImageNet.

**Actions Required To Increase Overall Recommendation:**

I strongly believe that the authors are tackling a problem that should be addressed, and because of this I am very willing to increase my score if there is additional work to demonstrate that this effect is helpful for larger architectures, more analysis is done on the choice of hyper-parameters, and a comparison to more recent methods is shown that the adaptive regularization regime for these methods is. a competitive.

**Clarity:**

The paper is written in a clean manner that is intuitive and easy to understand.

**Overall Review:**

Pros:
+ The work introduces a straightforward method for introducing adaptive task regularization in the class-incremental setting which is an understudied area that has an opportunity to be very impactful.
+ The authors make an effort to study the consistent effects of multiple datasets, something that is important for making larger claims.

Cons:
- It seems like there is no mention of the field of Continual Learning in the related works section of this paper despite the fact that it looks massively applicable to the work.
- Some of the experimental choices (size of lambda search range, number of samples, model architecture) don't seem to be very well motivated and arbitrary. I think that this work become much stronger if there were more ablations on modeling decisions and more results to answer the questions introduced at the beginning of the analysis.
- The chosen baselines with which to consider for regularization are both slightly older (2017) than some recent methods which might be more convincing. In general, it would probably be important for such an analysis to be broader over methods in class-incremental learning as there are many which claim to be significantly better than EWC and LwF.

**Potential Impact On The Field Of Automl:**

The setting of class-incremental learning is an instrumental one to AutoML as efficient methods that adapt to new tasks must be flexible to the introduction of new tasks. If using an adaptive regularization strategy is effective for producing models which are both better at new tasks as well as ensuring backward transferability, then it has large implications for how class-incremental learning approaches should optimize going forward. I think this paradigm stands to have a large impact in AutoML, but this work in particular because of its limited amount of analysis and lack of comparison to works might make it less impactful.

**Review Confidence:**

4: You are confident in your assessment, but not absolutely certain. It is unlikely, but not impossible, that you did not understand some parts of the submission or that you are unfamiliar with some pieces of related work.

**Review Rating:**

5: Borderline Leaning Reject: Technically sound paper where reasons to reject nonetheless outweigh reasons to accept. Please use sparingly.

**Review Summary:**

My current evaluation of this paper is Weak Reject. It is clearly written and tackles an interesting problem in adaptive task regularization but from my perspective struggles on two fronts. From a method perspective, the motivation as written at the end at the beginning of section 3 is not very well supported. From an experimental perspective, while the adaptive scheme shows clear improvement over some existing older methods in class-incremental learning, from my perspective, the evaluation is not very thorough either from considering an ablation of its parameters or how this task-adaptive regularization would compare to more recent class-incremental works.

**Technical Quality And Correctness:**

While the paper seems generally technically solid, there were a few points where I thought the procedure could be slightly more rigorous.

First, it seems that the models chosen for evaluation might be slightly on the smaller side of modern machine learning methods (ResNet32) where many classification works will usually use ResNet50.

Second, one of the drawn conclusions in the results section is "BWT performance significantly improved on EWC compared to LwF which signals EWC is more prone to forgetting, thus requires better adaptation to the current learning task." I don't know how you can necessarily draw this conclusion when EWC has a search space for lambda which was 2000x larger; perhaps it might be the case that there just wasn't a larger enough space explored. Another seemingly unusual parameter setting is the choice of weight decay going from 5e4 for the first task to 2e4 for the rest of the tasks.  In general, I feel that these ranges are probably crucial for this method, along with the number of uniform samples from them, and think the work would stronger with an ablation over these parameters.

Third, Figure 3 is an interesting demonstration of the variability of the regularization parameter; however, this chart is definitely highly variable when considering the relative order of tasks, something that isn't communicated.

Finally, there is an argument that is drawn at the end of section 3 regarding Low/High Plasticity and Low/High Stability, where the intuition is not very well supported for these claims. The human intuition of difficulty from learning something from a new semantic group probably doesn't hold here (a citation would strengthen this if it is indeed the case).

---

> ### Author Response · Authors · 2023-05-01
> **Respond to Reviewer wAxZ**
>
> We thank the reviewer for their detailed review. We answer the questions below.
>
> Comment: It seems like there is no mention of the field of Continual Learning in the related works section of this paper despite the fact that it looks massively applicable to the work.
>
> Reply: In this paper, we focus directly on Class-Incremental Learning, which is a special case of continual learning. To make this more clear, we extended the Related work section and provided related content under the subtitle of Class-Incremental Learning.
>
> Comment: Some of the experimental choices don't seem to be very well motivated and arbitrary. I think that this work become much stronger if there were more ablations on modeling decisions and more results to answer the questions introduced at the beginning of the analysis.
>
> Reply:
> Model Architecture:
> Our model architecture was not chosen arbitrarily. The architectural backbone we selected is designed for classification tasks and was previously presented at CVPR (He et al., 2016). Moreover, in each incremental step, we learn 10 classes that do not require big models like ResNet50. This also informed our selection. Our goal is to investigate the effect of adaptive regularization in continual learning regardless of the backbone architecture.
>
> Lambda Search Range and Number of Samples:
> Also, these were not selected arbitrarily. We performed a sensitivity analysis with the CIFAR 10 dataset to set the range of the regularization hyperparameter. We have now included these sensitivity analysis results in our Appendix. In these analyses, we observed that the lambda hyperparameter works best between [1, 100000] for EWC and [1, 50] for LwF, which indicates that LwF is more sensitive to its regularization hyperparameter. This also supports our selection of the number of samples: based on our observations, altering the regularization strength within a small range (e.g: increasing regularization strength from 1 to 10) does not have any discernible impact on the accuracy results of the Elastic Weight Consolidation (EWC) algorithm. However, in the case of Learning without Forgetting (LwF), there is a substantial difference in accuracy results when the regularization strength is increased. As a result of these findings, we chose the specified number of samples for our analysis. We have updated our paper and added related information on Appendix with respect to your comment.
>
> Comment: The chosen baselines with which to consider for regularization are both slightly older (2017) than some recent methods which might be more convincing. In general, it would probably be important for such an analysis to be broader over methods in class-incremental learning as there are many which claim to be significantly better than EWC and LwF.
>
> Reply: While more complex techniques have recently been proposed, Elastic Weight Consolidation and Learning without Forgetting methods are still very strong baselines and among the most preferred and used techniques in the continual learning field. Even the most recently proposed SOTA methods compare themselves with these methods. When setting up the experiments, we considered also comparing against the very latest techniques, but these all perform a combination of both regularization and memory-based techniques. In this paper, we wanted to directly study the effectiveness of using AutoML for adaptive regularization, and hence we focused on well-known pure regularization methods in continual learning, so that we could clerly analyse the results. Including memory-based mechanisms in automated continual learning is certainly part of our future research, but for this study it would inhibit a clear analysis of adaptive regularization itself. Given the range of existing memory-based mechanisms, this would require a whole new study to arrive at clear conclusions. As we mentioned in our limitation and conclusion sections, this work demonstrates the effectives of AutoML for automating regularization by itself, and we hope to inspire much more future research in auto-tuning the hand-crafted parameters in incremental learning.
>
> Comment: Figure 3 is an interesting demonstration of the variability of the regularization parameter; however, this chart is definitely highly variable when considering the relative order of tasks, something that isn't communicated.
>
> Reply:  Thank you! Indeed in the previous version of our paper, we presented results on a single. In the new version of our paper, we plotted the confidence intervals of the selected regularization strength values for 3 different seeds in Figure 3.

---

> > ### Author Response · Authors · 2023-05-01
> > **Rest of our Respond to Reviewer wAxZ**
> >
> > Comment: Finally, there is an argument that is drawn at the end of section 3 regarding Low/High Plasticity and Low/High Stability, where the intuition is not very well supported for these claims. The human intuition of difficulty from learning something from a new semantic group probably doesn't hold here.
> >
> > Reply:  The related citation was given in the mentioned section. In the referenced paper, the authors explain the “High Road Transfer” and “Low Road Transfer” concepts, where they similarly refer to the low/high stability/plasticity dilemma. Low road transfer is similar to low plasticity and high stability and it happens when the context of the new transfer is sufficiently similar to the prior context and accomplishing the new task is easily adopted. High road transfer in contrast is similar to low stability high plasticity dilemma where learning the new task demands more time and learning.

---

> > > ### Comment · Reviewer_wAxZ · 2023-05-01
> > > **Thank you for your response!**
> > >
> > > Thank you for the considerate discussion. After reading your response I agree that some parts which I thought were arbitrary definitely had thought and because of this I am going to raise my score from a 4 to a 5. However, there are still a few points left for clarification that I think is stopping me from raising my score further.
> > >
> > > RE: Model Architecture: I understand that the choice of ResNet is a decent one for the particular classification problem. However, when statements like "the effect of adaptive regularization in continual learning regardless of the backbone architecture." then I believe you need to consider a larger range of architectures to really ablate over what happens at different scales.
> > >
> > > RE: Lambda Search Range and Number of Samples. I strongly appreciate this, with this it makes much more sense to me why certain values were chosen.
> > >
> > > RE More complex techniques. This might be simply a matter of me not knowing the literature as well, but could you supply citations of recent SOTA methods which employ continue to employ LwF and EWC as their main baselines? Additionally, I understand the experimental setup kind of requires focusing on regularization-only techniques to evaluate the effect; however, in practice where people use techniques that combine with memory-based mechanisms more often, I think the paper would be stronger if the analysis extended to modern techniques which people might more often employ.

---

### Official Review · Reviewer_Cbki · 2023-04-13

**Potential Impact On The Field Of Automl Rating:** 3
**Technical Quality And Correctness Rating:** 2
**Clarity:** It is generally clear.
**Clarity Rating:** 3
**Actions Required To Increase Overall Recommendation:** Please answer the questions raised ab…

**Summary Of Contributions:**

This paper suggests a new adaptive regularization method in class-incremental learning.  Using Bayesian optimization, a coefficient for regularization is determined where a validation set is available.  The authors employ the tree-structured Parzen estimator as a Bayesian optimization strategy.  Finally, the experimental results of the proposed method prove that the proposed method is effective in class-incremental learning.

**Overall Review:**

* Frankly, I am not an expert of class-incremental learning, so it is somewhat difficult to properly evaluate the contributions of this paper to me.

* To my knowledge, the scale of experiments seems relatively small and baselines seem quite outdated.  According to Section 6, the proposed method can be easily adopted in any models in a plug-and-play manner, but I think it should be verified by numerical analyses.  Although the contributions of class-incremental learning seem limited, I do not degrade the evaluations of this paper now.  I would like to listen to the authors' and other reviewers' thoughts.

* I would like to mainly leave a comment on Bayesian optimization part.  Since this work only optimizes a single coefficient, I am not sure that Bayesian optimization is an appropriate choice.  Did you try other techniques like simple derivative-free algorithms or genetic algorithms?

* Moreover, I think that the tree-structured Parzen estimator is okay, but you need to attempt to use other methods in Bayesian optimization.  Using the RayTune library, you can easily try other techniques with the modification of a few lines of code.

* Also, I am curious about the property of a validation dataset.  I think that experimental results are surprising, because this method is very effective.  Is there any assumptions that the validation dataset is similar to future tasks?  If the validation and future datasets are significantly different, do you think the proposed method still works?

**Potential Impact On The Field Of Automl:**

Since this paper demonstrates an interesting application of Bayesian optimization, it has a potential impact on the field of AutoML.

**Review Confidence:**

2: You are willing to defend your assessment, but it is quite likely that you did not understand the central parts of the submission or that you are unfamiliar with some pieces of related work.

**Review Rating:**

5: Borderline Leaning Reject: Technically sound paper where reasons to reject nonetheless outweigh reasons to accept. Please use sparingly.

**Review Summary:**

Please see the text boxes above.

**Technical Quality And Correctness:**

It seems correct, but the scale of experiments is small and baselines are outdated.  Also, the Bayesian optimization part should be investigated more.

---

> ### Author Response · Authors · 2023-05-01
> **Respond to Reviewer Cbki**
>
> We thank the reviewer for their detailed review. We answer the questions below.
>
> Comment: To my knowledge, the scale of experiments seems relatively small and baselines seem quite outdated. According to Section 6, the proposed method can be easily adopted in any models in a plug-and-play manner, but I think it should be verified by numerical analyses. Although the contributions of class-incremental learning seem limited, I do not degrade the evaluations of this paper now.
>
> Reply: While more complex techniques have recently been proposed, Elastic Weight Consolidation and Learning without Forgetting methods are still very strong baselines and among the most preferred and used techniques in the continual learning field. Even the most recently proposed SOTA methods compare themselves with these methods. When setting up the experiments, we considered also comparing against the very latest techniques, but these all perform a combination of both regularization and memory-based techniques. In this paper, we wanted to directly study the effectiveness of using AutoML for adaptive regularization, and hence we focused on well-known pure regularization methods in continual learning, so that we could clearly analyze the results. Including memory-based mechanisms in automated continual learning is certainly part of our future research, but for this study it would inhibit a clear analysis of adaptive regularization itself. Given the range of existing memory-based mechanisms, this would require a whole new study to arrive at clear conclusions. As we mentioned in our limitation and conclusion sections, this work demonstrates the effectives of AutoML for automating regularization by itself, and we hope to inspire much more future research in auto-tuning the hand-crafted parameters in incremental learning.
>
> Comment: I am not sure that Bayesian optimization is an appropriate choice. Did you try other techniques like simple derivative-free algorithms or genetic algorithms?
>
> Reply: Our early analysis of these regularization parameters showed that the response surface is smooth, but also quite complex - it is very hard to predict this surface for any given current (source) and new (target) task. Although we didn’t directly evaluate derivative-free algorithms, we expect them to be quite suboptimal given the complexity of the response surface. Bayesian Optimization is a probabilistic approach that efficiently explores the most promising regions of the hyperparameter space, and can therefore adapt very efficiently to the response of the model when encountering a new task. It has been shown to work very well when the objective function is smooth and continuous, and the hyperparameters are few and continuous or categorical. On the other hand, genetic search is much less efficient in this particular setting, e.g. it would have to go through many crossover and random mutation operators across many generations. This is better suited for high-dimensional search spaces, but we expected it to be much less efficient in this particular setting. Since we are optimizing towards a single hyperparameter (regularization strength) we selected Bayesian optimization in our study, and it works well. Another reason we selected Bayesian optimization is that it opens the door to a range of meta-learning approaches (e.g. warm-starting and surrogate model transfer) which could be explored in future work to learn across tasks and adapt models even faster.
> Comment: Moreover, I think that the tree-structured Parzen estimator is okay, but you need to attempt to use other methods in Bayesian optimization. Using the RayTune library, you can easily try other techniques with the modification of a few lines of code.
> Reply: This is indeed a very interesting aspect to investigate in future work, and we would be happy to present more results in later versions of this paper. Since we focused on presenting the effectiveness of the adaptive regularization in class incremental scenarios, and TPE is known to perform very well given our search space, we experimented with this first. Moreover, it is a very scalable and relatively simple approach that could be adopted easily by the continual learning community. It is certainly possible that other techniques work better, and we explore this in future work.
>
> Comment:  Is there any assumptions that the validation dataset is similar to future tasks? If the validation and future datasets are significantly different, do you think the proposed method still works?
>
> Reply: The validation set does not include any future tasks: For example, if we are training on Task 3, our validation set consists of 20% of Task 1 + 20% of Task 2 + 20% of Task 3 to select the best regularization strength. Thus, our validation set includes old tasks to ‘remember’ and new tasks to ‘learn’ by selecting the best regularization strength. Hence, we are sure that adaptive regularization would still help even if validation and future datasets are different.

---

> > ### Comment · Reviewer_Cbki · 2023-05-02
> > **Thank you for your response**
> >
> > Thank you for your response.
> >
> > I acknowledge that I have read your rebuttal.  After reading it, I would like to maintain my score.